# FlashDLM: Accelerating Diffusion Language Model Inference via Efficient KV Caching and Guided Diffusion

**Zhanqiu Hu**[1*]  **Jian Meng**[1,2*]  **Yash Akhauri**[1]
**Mohamed S. Abdelfattah**[1]  **Jae-sun Seo**[1]  **Zhiru Zhang**[1]  **Udit Gupta**[1]
[1]Cornell University  [2]Huawei
[*]Equal contribution

## Abstract

Diffusion language models offer parallel token generation and inherent bidirectionality, promising more efficient and powerful sequence modeling compared to autoregressive approaches. However, state-of-the-art diffusion models (e.g., Dream 7B, LLaDA 8B) suffer from slow inference. While they match the quality of similarly sized Autoregressive (AR) Models (e.g., Qwen2.5 7B, Llama3 8B), their iterative denoising requires multiple full-sequence forward passes, resulting in high computational costs and latency, particularly for long input prompts and long-context scenarios. Furthermore, parallel token generation introduces token incoherence problems, and current sampling heuristics suffer from significant quality drops with decreasing denoising steps. We address these limitations with two training-free techniques. First, we propose `FreeCache`, a Key-Value (KV) approximation caching technique that reuses stable KV projections across denoising steps, effectively reducing the computational cost of DLM inference. Second, we introduce `Guided Diffusion`, a training-free method that uses a lightweight pretrained autoregressive model to supervise token unmasking, dramatically reducing the total number of denoising iterations without sacrificing quality. We conduct extensive evaluations on open-source reasoning benchmarks, and our combined methods deliver an average of **12.14×** end-to-end speedup across various tasks with negligible accuracy degradation. For the first time, diffusion language models achieve a comparable and even faster latency as the widely adopted autoregressive models. Our work successfully paved the way for scaling up the diffusion language model to a broader scope of applications across different domains. Our code and implementation are available at `https://github.com/ZhanqiuHu/flash-dlm-experimental`.

## 1 Introduction

Large language model (LLM) Liu et al. (2024a); Bai et al. (2023); Touvron et al. (2023) have shown great success across multiple domains in human life with strong intelligence. In particular, most of the current state-of-the-art (SoTA) LLM models are autoregressive (AR) models with transformer-based architecture Vaswani et al. (2017). However, autoregressive models sequentially generate new tokens at inference time for each decoding step, preserving the logic of semantics (e.g., logic) with the cost of insufficient parallelism and the limitation of mono-direction generation. Diffusion Language Models (DLMs) Ye et al. (2025); Nie et al. (2025); Arriola et al. (2025) have emerged as an appealing alternative. By iteratively denoising a masked sequence in parallel, DLM enables parallel token generation and bidirectional context, which allows all tokens to be generated simultaneously at each step, while keeping the coherence between past and future tokens.

**However, DLMs exhibit fundamental challenges that remain under-explored.** The comparable quality between the SoTA DLM and AR model comes with the cost of extensive computation latency and insufficient *long-context* ability. These drawbacks largely limit the scalability and practicality of diffusion models, which motivates this work to investigate the following problem:

*How to effectively accelerate SoTA DLM while maintaining model performance?*

Essentially, the efficiency bottleneck of DLM is caused by the incompatibility with the caching strategy, which has been widely used in auto regressive models (ARM). Recently proposed Block Diffusion Arriola et al. (2025) have shown promising results on diffusion model with Key-Value caching strategy, whereas the overhead of additional fine-tuning and training hinder the fast deployment of DLM. More importantly, the scalability of caching remains questionable against large-sized DLMs (e.g., Dream-7B-Instruct Ye et al. (2025), LLaDA-8B-Instruct Nie et al. (2025))

To address the efficiency challenge of the SoTA DLM models, this work proposes two novel methods: ① `FreeCache`, a training-free caching strategy that enables diffusion acceleration; ② `Guided Diffusion`, a training-free technique that guides token unmasking in long-context diffusion generation. Unlike the conventional compression algorithm (e.g., quantization, pruning) that requires a dedicated calibration run, the proposed method directly accelerate the model off the shelf. Specifically, we reveal the fact that the impact of future tokens on earlier positions rapidly diminishes over denoising steps. Therefore, direct caching of the earlier unmasked clean tokens can be considered as a KV state approximation with the result of negligible impact on output quality.

In addition to FreeCache, we introduce guided unmasking assisted by a lightweight autoregressive model. The proposed Guided Diffusion introduces the mechanism of "big diffusion model generation followed by small model unmasking guidance." Our guidance-only approach selects plausible tokens to unmask, minimizing this overhead and preserving the full reasoning power of the DLM. By doing so, the proposed method elegantly combines the advantages of both DLM and autoregressive models. The diffusion process can be easily parallelized, while the autoregressive model is **exempt** from repetitive and expensive speculative correction. Overall, the contributions of this work are:

- **Simplicity.** The proposed method achieves an average of 12.14× and 13.29× speedup on the Dream-7B-Instruct and LLaDA-8B-Instruct models with negligible accuracy drop, respectively. For the first time, the DLM based architecture achieves comparable (even better) generation speed compared to the same-sized AR-based LLM models.

- **Versatility.** The proposed method exhibits strong performance across multiple knowledge domains, including both simple and complex reasoning tasks.

- **Scalability.** With the proposed training-free auto-regressive guidance, our work enables the long-context diffusion (>1024 tokens) **without** hurting the model performance.

Together, these techniques make it feasible to deploy DLMs in long-context and high-throughput applications without compromising quality. Our method retains the desirable properties of parallel generation and bidirectional conditioning of DLMs while overcoming the challenge of key inefficiencies. We demonstrate that diffusion-based language models can achieve competitive inference performance with autoregressive LLM baselines, paving the way for their practical adoption in real-world scenarios.

## 2 RELATED WORK

### 2.1 DIFFUSION LANGUAGE MODELS

The emergence of the diffusion language model (DLM) opens up an alternative path for language generation. Recent works like Dream Ye et al. (2025), LLaDA Nie et al. (2025), MDLM Sahoo et al. (2024), and BD3-LM Arriola et al. (2025) adapt diffusion processes to text generation. While promising, these methods are compute-intensive because iterative denoising requires repeated full-sequence forward passes (see Section 3.1.1), leading to substantially higher latency compared to conventional AR-based language models despite the high scalability. With some preliminary trials, recent work imposes cached diffusion models Arriola et al. (2025) for accelerated diffusion and reinforcement learning-based reasoning enhancement Zhao et al. (2025). However, the fundamental latency issue still persists, which largely limits the practicality and scalability of DLM. More importantly, diffusion and autoregressive-based language models are not orthogonal. The potential compatibility between DLM and Autoregressive Model is worth further exploration. Earlier foundations in discrete diffusion include D3PM Austin et al. (2023), which introduced structured transition matrices for discrete state

spaces, SEDD Lou et al. (2024), which models score ratios for discrete diffusion, and MD4 Shi et al. (2025), which simplifies and generalizes masked diffusion for discrete data.

Several recent methods have also been proposed to accelerate DLM inference. dKV-Cache Ma et al. (2025) reuses intermediate KV states with heuristic scheduling decisions such as delay offsets, window sizes, and periodic refresh steps. Fast-dLLM Wu et al. (2025) introduces confidence-driven parallel decoding rules that require task-specific hyperparameter tuning. In contrast, our approach avoids heuristic scheduling and threshold tuning entirely: FreeCache freezes KV projections of finalized blocks without recomputation, and Guided Diffusion uses model agreement as the sole unmasking signal.

## 2.2 KV Caching for Autoregressive LLM

The overhead of KV Cache has been one of the most critical memory and computation overheads throughout the long-context generation with autoregressive LLM decoding. Compressing the KV cache has been one of the major focuses of efficient LLM research. The emergence of FlashAttention Dao et al. (2022); Dao (2023); Shah et al. (2024) and PageAttention Kwon et al. (2023) aims to alleviate the challenge from the system perspective, leading to a promising performance with high scalability across different LLM architectures. From the perspective of compression algorithms, KV cache eviction Yang et al. (2024); Cai et al. (2024b), quantization Liu et al. (2024c); Hooper et al. (2024), structured pruning Lu et al. (2024); Xu et al. (2025) are proposed to save memory traffic. Due to the irregularity of the diffusion process, the diffusion language model (DLM) requires the full-sized Key and Values for attention computation, where the bottleneck of memory bound largely degrades the latency, limiting the practicality of DLM on a larger scale. A recent study on block diffusion Arriola et al. (2025) proposes the sequential block with intra-block diffusion, whereas the cost of training overhead and under-verified scalability makes the diffusion model questionable for large-scale deployment.

## 2.3 Speculative Decoding

In addition to the conventional compression algorithms (e.g., pruning and quantization), speculative decoding Leviathan et al. (2023) provides an alternative perspective on accelerating autoregressive LLM decoding without hurting the performance (e.g., accuracy). The lightweight assistant model generates a multi-token "draft", followed by the speculation and correction of the target model. The reliability and acceleration of the speculative decoding system is determined by the matching ratio between the draft model output and target model speculation. In particular, the poor matching ratio between the draft and target model leads to frequent execution of the target model, which eventually deteriorates the system level latency. Recent studies on autoregressive models focus on self-drafting Elhoushi et al. (2024); Liu et al. (2024b), multi-head decoding Cai et al. (2024a), optimized token sampling Chen et al. (2024) and search strategy. Regardless the design of speculative decoding, the speculation and correction of the target model is actively required to ensure the lossless computation. In the meantime, the diffusion model-based big-small model collaboration is largely under investigation. Recent work Christopher et al. (2024) explores speculative decoding with small-scale diffusion models as the drafter, relying on a more competent AR model for output quality and generation, whereas cross-model methods for accelerating large, pre-trained diffusion language models remain under-explored.

## 3 Methodology

### 3.1 Preliminary of Diffusion Language Model

Diffusion language models (DLMs) generate sequences by iteratively unmasking an initially masked token sequence over $T$ denoising steps. Mathematically, let $\mathbf{x}_T \in \mathbb{N}^L$ be the fully masked input (all positions set to [MASK]), where the still masked position $M_t$ is represented as:

$$M_t = \big\{\, i : x_t[i] = \text{[MASK]} \,\big\} \tag{1}$$

At each iteration $t = T, \ldots, 1$, the model $f_\theta$ produces logits

$$z_t = f_\theta(\mathbf{x}_t) \quad \text{for } i \in M_t, \tag{2}$$

from which a subset $U_t \subseteq M_t$ of positions is selected (e.g. by confidence scoring or at random) and filled via:

$$\mathbf{x}_{t-1}[i] = \pi\big(\text{Softmax}(z_t[i])\big), \quad i \in U_t, \tag{3}$$

where $\pi$ denotes the token selection policy (e.g., greedy $\arg\max$, temperature sampling, or top-$k$).

yielding the next mask set $M_{t-1} = M_t \setminus U_t$. By repeating this denoising and unmasking process until $M_0 = \emptyset$, the model produces the final fully unmasked sequence $\mathbf{x}_0$.

| Module | Operation | Auto-Regressive LM (decode) | Diffusion LM (length $L$) |
|--------|-----------|:---------------------------:|:-------------------------:|
| MHA | $W_Q, W_K, W_V$ projections | $\mathcal{O}(d^2)$ | $\mathcal{O}(Ld^2)$ |
| | Query $\times$ Key ($QK^\top$) | $\mathcal{O}(l^2 d/h)$ | $\mathcal{O}(L^2 d/h)$ |
| | Attention Score $\times$ Value | $\mathcal{O}(ld/h)$ | $\mathcal{O}(L^2 d/h)$ |
| | Output projection ($W_{\text{out}}$) | $\mathcal{O}(d^2)$ | $\mathcal{O}(Ld^2)$ |
| FFN | $W_1$ projection | $\mathcal{O}(dd_{\text{ff}})$ | $\mathcal{O}(Ldd_{\text{ff}})$ |
| | $W_2$ projection | $\mathcal{O}(dd_{\text{ff}})$ | $\mathcal{O}(Ldd_{\text{ff}})$ |

Table 1: Compute complexity of Transformer modules: AR decodes a prefix of length $l$ per token vs. DLM processes full length $L$ each denoising step.

### 3.1.1 BOTTLENECK OF DIFFUSION LANGUAGE MODEL INFERENCE

While diffusion models allow parallelized generation of tokens, this comes at the expense of extra computation overhead per forward pass. Given the input $x_T$ with sequence length $L$, each forward pass of the diffusion model $f_\theta$ requires full-sized computation (both current and past tokens) throughout the multi-head attention (MHA) block and feed-forward network (FFN) of the transformer model. Unlike the autoregressive model where the past tokens are cached for the new token generation, diffusion model produces output without the consideration of causality. As shown in Table 1, DLM models introduce an additional computational complexity of $O(L)$ at each module of a transformer-based model architecture. As a result, the latency overhead introduced by the DLM architecture is massive compared to the Auto-Regressive Model, as shown in the comparison in Section 4. Based on the theoretical analysis above, the first challenge of the diffusion model is:

**Challenge①**: *How to alleviate the latency and computation overhead caused by the liability of full sized cache computation of DLM?*

### 3.1.2 TOKEN INCOHERENCE IN PARALLEL GENERATION

As introduced in Section 3.1, DLM irregularly unmasks output token positions throughout diffusion process. As a result, simultaneously unmasking token positions without semantic correlation leads to insufficient semantic consistency, which can further degrades the model performance. Fundamentally, parallel diffusion decoding relies on a factorized distribution over masked positions, which limits its ability to model joint dependencies during parallel unmasking. While the desire of preserving the parallel diffusion persists. Figure 1a shows accuracy degradation with decreasing denoising steps for three diffusion sampling methods: MaskGIT Chang et al. (2022), entropy-based, and Top-K Margin Kim et al. (2025). The challenge arises as follows:

**Challenge②** *How to enhance the semantic correlation of the diffusion process **without** introducing additional training and fine-tuning effort?*

Motivated by the aforementioned challenges, we propose a novel approach which directly accelerates the diffusion model in a simple, effective, and systematic manner. Our proposed Guided Diffusion mitigates this factorization limitation by requiring agreement between the DLM and an AR model before committing multiple positions, effectively reducing incoherence from independent token predictions without altering the underlying diffusion process.

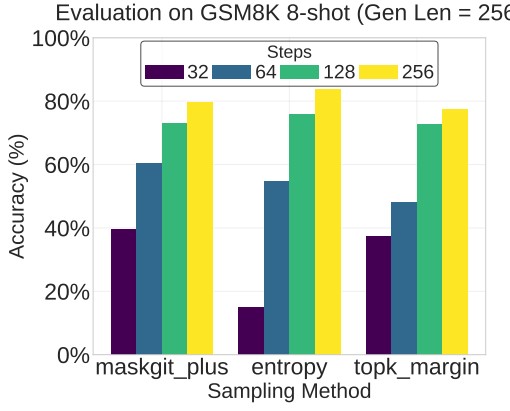 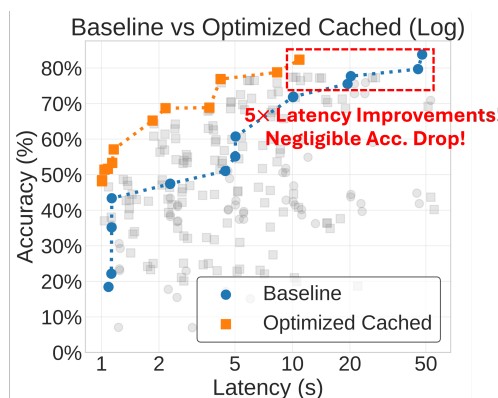

(a) **Quality degradation from aggressive unmasking.** Heuristic methods suffer from significant accuracy loss with decreasing denoising steps.

(b) **Latency-accuracy tradeoff.** Pareto front comparing the standard autoregressive baseline with our KV-cache optimized method, highlighting substantial latency reductions at equivalent accuracy levels.

Figure 1: **(a)** Quality degradation under aggressive unmasking. **(b)** Pareto front of accuracy vs. latency (log-scale) for baseline and optimized cached methods.

### 3.2 PROPOSED METHOD

#### 3.2.1 FREECACHE: REDUCING WINDOW CACHING FOR DLM INFERENCE

A key insight for optimizing the generation process in Diffusion Language Models is that the Key and Value (KV) projections for the clean portions of a sequence exhibit high temporal stability. Although these projections are not strictly unchanging, they quickly converge and undergo only negligible changes once a token is finalized.

Figure 2 shows a heatmap that validates the rationale behind our FreeCache method: the temporal stability of KV projections for clean tokens. The y-axis represents the denoising step and the x-axis represents the token position, while the color intensity shows the cosine similarity of a token's value projection to its previous step. Bright yellow indicates high similarity (minimal change), while dark blue signifies a large difference. We observed that the prompt region remains stable across denoising steps, and the generation region illustrates how tokens transition from a low-similarity, unstable state (dark blue) to a high-similarity, stable state (bright yellow) as they become unmasked.

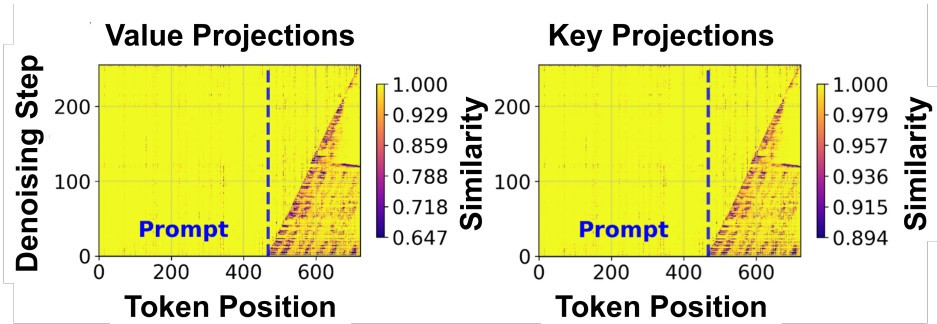

Figure 2: Rationale behind the proposed FreeCache: The variation of K and V projections of clean tokens is small throughout the subsequent diffusion steps (measured via cosine similarity).

To exploit this, we propose `FreeCache`, a reducing window caching strategy where the set of actively computed tokens dynamically shrinks as blocks of the sequence are finalized. The algorithm proceeds as follows: **Initialization and Partitioning:** The post-prompt generation sequence is

Table 2: Latency and accuracy of the proposed FreeCache with different Diffusion Language Models. The proposed method achieves $4.42\times$ and $6.32\times$ speedup on Dream-7B-Instruct and LLaDA-8B-Instruct models.

| Models | Method | GSM8K (8-shot) Accuracy (%) | Raw Latency (s) | Speedup |
|---|---|---|---|---|
| Dream-7B-Instruct Ye et al. (2025) | Baseline | 79.68 | 48.05 | $1.0\times$ |
| | **FreeCache (This work)** | 77.40 | **10.87** | **4.42$\times$** |
| LLaDA-8B-Instruct Nie et al. (2025) | Baseline | 79.30 | 56.89 | $1.0\times$ |
| | **FreeCache (This work)** | 77.18 | **9.02** | **6.32$\times$** |

partitioned into fixed-size blocks $(B_1, ..., B_N)$, and an initial forward pass computes and saves the full KV projections for the entire sequence (prompt + all blocks). **Windowed Re-computation:** To generate a block $B_i$, the **active computation window** is defined as $B_i$ and all subsequent blocks $(B_{i+1}, ..., B_N)$. KV projections are recomputed only for tokens within this window until $B_i$ is fully unmasked, using all prior frozen blocks and the prompt as context. **Progressive Caching:** Upon completion of a block $B_i$, its KV projections are **frozen** for subsequent steps. The active window then shrinks to exclude this block for all subsequent generation steps.

This cascaded approach means that the computational cost per step progressively decreases throughout the inference process, as the active window shrinks with each completed block. This effectively reduces the overall computational overhead, especially for long sequences. While the current design uses a fixed block size for simplicity, FreeCache is compatible with dynamic block-size policies. Potential extensions include adapting the block size based on local KV stability, diffusion uncertainty, or the AR-agreement rate from Guided Diffusion. Longer-term directions include cache-aware or schedule-aware training so that the model learns KV dynamics that better support caching.

We quantify the performance degradation caused by the `FreeCache` in Figure 1b with the 8-shot GSM8K dataset. The reducing window caching strategy introduces up to $5\times$ speedup compared to the vanilla Dream-7B, while preserving the minimal accuracy drop. As shown in Table 2, the proposed FreeCache achieves up to $4.42\times$ speedup for Dream-7B-Instruct and $6.32\times$ for LLaDA-8B-Instruct, while maintaining accuracy close to the baseline. We solidify the performance of the proposed method in Section 4 with further analysis with benchmarks across different domains.

### 3.2.2 GUIDING DIFFUSION WITH LIGHTWEIGHT AUTOREGRESSIVE MODEL

To address this token incoherence challenge (Challenge ②) from Section 3.1.2 and fully exploit the parallelism of diffusion process, we propose `Guided Diffusion`.

Specifically, we leverage the diffusion model as the "drafter" model by first unmasking **all** the tokens with one step. The masked tokens predicted by the diffusion model are only unmasked when they are consistent with the predictions from the autoregressive LM model, as shown in Figure 3. For the scenario when the matching ratio between the diffusion drafter and auto-regressive model is zero, the guided diffusion will only accept the one free tokens from the AR model output.

This cross-model agreement serves as a lightweight coherence prior, enabling the model to safely unmask multiple tokens in parallel, without requiring any additional training or fine-tuning.

`Guided diffusion` performs iterative unmasking by coordinating predictions from a diffusion language model (DLM) and a frozen autoregressive model (ARM). At each denoising step, instead of relying on heuristic confidence thresholds, it uses agreement between the two models to decide how many tokens to safely unmask.

Formally, let $x_T \in \mathbb{N}^L$ be the current sequence containing masked and unmasked tokens. The diffusion model $f_\theta$ predicts logits over vocabulary tokens at all masked positions, from which tokens are selected via policy $\pi$:

$$t^{\text{DLM}} = \pi(\text{Softmax}(f_\theta(x))).$$

These proposed tokens are fed to the ARM $g_\phi$, which produces a set of logits:

$$\text{logits}^{\text{AR}} = \text{Softmax}(g_\phi(t^{\text{DLM}})).$$

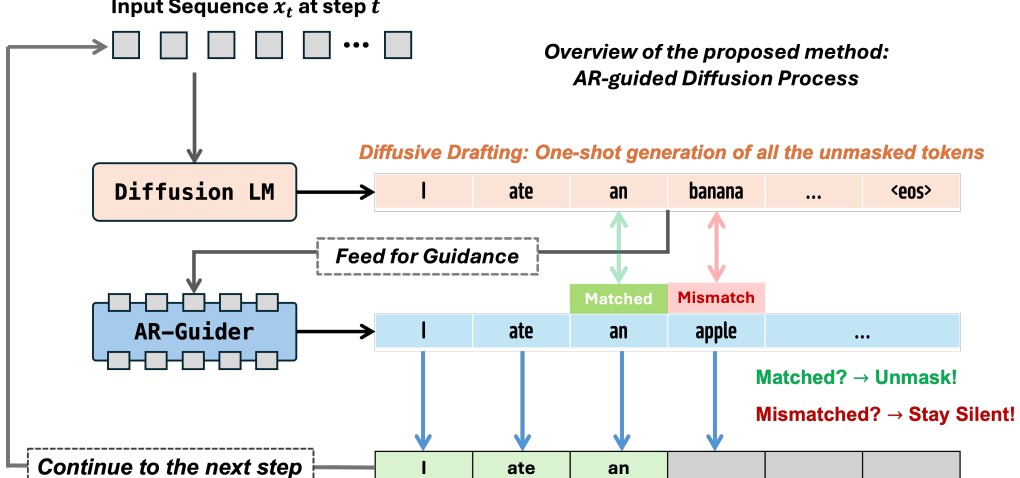

Figure 3: AR-guided Diffusion Model. The diffusion model performs a one-step diffusion process, followed by a one-time forward pass of the AR guider. The matched tokens are unmasked for the next step of diffusion.

Let $M = [i_1, \ldots, i_m] \subseteq \{1, \ldots, L\}$ denote the current masked positions in $x$. We compute the longest agreed prefix length $k$ via matching policy $\mathcal{M}$:

$$k = \mathcal{M}(t^{\text{DLM}}, \text{logits}^{\text{AR}}).$$

If $k > 0$, we unmask positions $i_1, \ldots, i_k$ using the DLM proposals. If no agreement is found, we conservatively unmask only the first token $i_1$. This process is repeated until all tokens are unmasked.

This decoding strategy allows the model to adaptively control the amount of parallel unmasking based on model confidence and alignment, enabling efficient generation without compromising output quality. We present the details of the proposed method in Algorithm 1. In this work, we support both exact match and a relaxed variant that accepts near-ties via a confidence threshold $\tau$ (see Section C.4 for details). Other matching policies are also compatible with the framework.

We would like to highlight that the proposed `Guided Diffusion` properly combines the advantages of **both** diffusion model and auto-regressive model. For each guiding step, **both** one-time diffusion process from the drafter and the one-time forward pass of the auto-regressive model are fast, while the guidance from the auto-regressive model facilitates the coherence of the diffused output tokens with improved semantic logic.

**Guided Diffusion vs. Speculative Decoding.** In Speculative Diffusion Decoding (SDD) Christopher et al. (2024), the AR model serves as the target and the diffusion model acts as a drafter whose proposals may be corrected by the AR model. Unlike speculative decoding, `Guided Diffusion` does not require token-wise correction from the auto-regressive model. In other words, the proposed method produces output sequence **without** introducing the repetitive correction process, which guarantees the latency benefits obtained from the fast diffusion process. Speculative decoding uses a small AR model for autoregressive drafting, followed by the big model correction. On the contrary, the proposed guided diffusion flips the paradigm by maximizing the drafting efficiency of the big diffusion model. Because the AR guider only provides an agreement signal rather than generating or correcting tokens, a small, lightweight AR model suffices for this role, minimizing the additional latency overhead. In addition, the reasoning power of the diffusion model is fully preserved regardless of the choice of the AR guider, as supported empirically in Table 4.

## 4 EXPERIMENTS

We evaluated the proposed method against a wide range of benchmarks across different types of tasks. Due to the limited option of DLM models with multi-billion parameters, we choose Dream-

**Algorithm 1:** Guided Diffusion

**Input:** masked sequence $x$, DLM $f_\theta$, AR $g_\phi$, token selection policy $\pi$, matching policy $\mathcal{M}$
**Output:** fully unmasked $x$
**while** $\exists i : x[i] = \textit{mask}$                // loop until all unmasked **do**
    |  $\text{logits}_{\text{DLM}} \leftarrow \text{softmax}(f_\theta(x))$         // DLM predicts masked tokens
    |  $\text{tok}_{\text{DLM}} \leftarrow \pi(\text{logits}_{\text{DLM}})$                // DLM predictions
    |  $\text{logits}_{\text{AR}} \leftarrow \text{softmax}(g_\phi(\text{tok}_{\text{DLM}}))$     // AR processes DLM output
    |  Let $M = [i_1, \ldots, i_m]$ be masked indices    // remaining mask positions
    |  $k \leftarrow \mathcal{M}(\text{tok}_{\text{DLM}}, \text{logits}_{\text{AR}})$             // find prefix matches
    |  **if** $k > 0$                            // models agree **then**
    |     |  Unmask $i_1{:}i_k$ with $\text{tok}_{\text{DLM}}[i_1{:}i_k]$     // accept matching run
    |  **else**
    |     |  Unmask $i_1$ with $\text{tok}_{\text{DLM}}[i_1]$          // fallback: accept first

$\pi$: token selection policy (e.g., greedy, temperature, top-$k$).
$\mathcal{M}$: matching policy that returns the length of the longest agreed prefix; e.g., exact match where $\text{tok}_{\text{DLM}}[i]$ equals the top-1 token of $\text{logits}_{\text{AR}}[i]$.

Table 3: Outstanding acceleration achieved by the proposed methods with negligible accuracy drop on Dream-7B-Instruct Ye et al. (2025). The latency value represents the end-to-end problem solving time per problem.

| | | Vanilla DLM | Dream-Instruct-7B with Guided Diffusion (This Work) | | |
|---|---|---|---|---|---|
| | | Baseline Dream-Instruct-7B | FreeCache Only | FreeCache + QWen-1.5B-Instruct | FreeCache + QWen-7B-Instruct |
| **MMLU-PRO** | Acc. (%) | 46.92 | 45.18 | 46.64 | 48.20 |
| | Latency (s) | 20.73 | **6.68** | **1.66** | **2.77** |
| | Speedup (s) | 1× | **3.11×** | **12.48×** | **7.48×** |
| **GSM8K (8-shot)** | Acc. (%) | 79.68 | 77.40 | 80.33 | 81.41 |
| | Latency (s) | 48.05 | **10.87** | **2.70** | **2.74** |
| | Speedup | 1× | **4.42×** | **17.80×** | **17.53×** |
| **PiQA** | Acc. (%) | 85.56 | 84.83 | 85.15 | 85.85 |
| | Latency (s) | 14.62 | **4.21** | **0.43** | **1.05** |
| | Speedup | 1× | **3.47×** | **34.1×** | **13.92×** |
| **ARC-C** | Acc. (%) | 80.87 | 80.61 | 80.89 | 80.72 |
| | Latency (s) | 10.64 | **4.25** | **3.12** | **4.58** |
| | Speedup | 1× | **2.50×** | **3.41×** | **2.32×** |
| **ARC-E** | Acc. (%) | 87.53 | 86.24 | 87.50 | 87.44 |
| | Latency (s) | 10.59 | **5.54** | **3.25** | **5.03** |
| | Speedup | 1× | **1.91×** | **3.26×** | **2.11×** |
| **GPQA** | Acc. (%) | 39.29 | 43.30 | 49.55 | 49.33 |
| | Latency (s) | 21.50 | **9.91** | **12.02** | **15.26** |
| | Speedup | 1× | **2.12×** | **1.78×** | **1.41×** |

Instruct-7B Ye et al. (2025) as the selected SoTA model to validate the proposed method. To fully solidify the effectiveness of our work, both question-answer (QA) and reasoning tasks are solved via multi-sentence and multi-step text-based diffusion, instead of directly outputting the final answer. The detailed experimental setup and data pre-processing are summarized in the supplemantary.

**Metrics and Baselines.** The performance of the proposed method and other baselines are evaluated based on accuracy and wall clock latency for solving each problem. We first compare the performance of the proposed method against the vanilla Dream model Ye et al. (2025). On top of that, we evaluate the proposed method with the comparison of the mainstream auto-regressive models.

**Experimental Setup.** The models are deployed on a single NVIDIA RTX 6000 Ada GPU (48GB). The latency is measured directly via `torch.cuda.Event` across each benchmark dataset.

**Long-context Diffusion.** For all the tasks, we intentionally encourage the model to generate the complete reasoning process, followed by the indication of the final answer. We would like to highlight the long context response is critical for problem solving and question-answering due to the widely

adopted reasoning-based response in the SoTA LLM models. For all the selected reasoning, math problems, and question-answering, we set the `max_new_tokens` to 1024. The final answers are extracted after the dedicated prompt trigger.

## 4.1 COMPARISON WITH THE BASELINE DIFFUSION LANGUAGE MODEL

Compared to the vanilla Dream-7B-Instruct model, the proposed method achieves consistent and outstanding speedup across all the tasks from different domains. For the complex reasoning tasks (e.g., MMLU-PRO, GPQA), the proposed `FreeCache` first induces $3.11\times$ speedup with minimal quality degradation. Rewarded by the guided diffusion method, the overall system can achieve an average of $\mathbf{12.48\times}$ speedup with negligible accuracy degradation. For the tasks with long-context input prompt (e.g., GSM8K with 8-shot), the proposed method accelerates the diffusion process with $\mathbf{34.1\times}$ end-to-end average speedup on all the math problems, as presented in Table 3.

With guided unmasking, the diffusion model exhibits stronger reasoning power with recovered accuracy. More importantly, the existence of the lightweight autoregressive model recovers the accuracy degradation caused by the approximated cache, even with the long-context generation (e.g., 1024 tokens). As introduced in Section 1, the guided diffusion process exclude the "correction" step. As a result, increasing the size of the guider model (e.g., from 1.5B to 7B) only leads to marginal accuracy improvements, with the cost of minimal latency overhead.

Empowered by `Guided Diffusion`, diffusion model is compatible with different variants of the AR model guider in different domains of knowledge. By employing a dedicated auto-regressive LLM for math (e.g., Qwen-2.5-Math-1.5B), the performance of the Dream-Instruct-7B model can be further facilitated with better reasoning ability on the GSM8K dataset. In addition to the experimental results in Table 3, we present the evaluation results with other benchmarks in the supplementary.

## 4.2 COMPARISON WITH THE VANILLA AUTOREGRESSIVE LLM MODELS

The strong speedup achieved by the proposed method allows us to compare the diffusion model against the conventional autoregressive LLM models.

Table 4 shows the comparison between Guided Diffusion and vanilla Auto-regressive on the GSM8K dataset with the standard 8-shot chain-of-thought prompt. Crucially, the quality of generation is governed by the DLM's reasoning capacity. This ensures that accuracy is maintained regardless of guider competence, with latency remaining on par with autoregressive models. Admittedly, Dream-7B-Instruct and LLaDA-8B currently exhibit weaker reasoning ability compared to some state-of-the-art autoregressive models of similar size (e.g., Qwen2.5-7B-Instruct). Nevertheless, Guided Diffusion provides significant acceleration, and as future work continues to enhance the reasoning ability of diffusion models, its benefits are expected to become even more pronounced.

Table 4: Comprehensive performance comparison on the GSM8K dataset, showing both standard autoregressive (AR) and AR-guided results for various models.

| Models / Guiders | Stand-alone AR LLM | | Guided Diffusion (w/ Dream-7B-Instruct) | | Guided Diffusion (w/ LLaDA-8B-Instruct) | |
|---|---|---|---|---|---|---|
| | Accuracy (%) | Latency (s) | Accuracy (%) | Latency (s) | Accuracy (%) | Latency (s) |
| Qwen2.5-1.5B-Instruct | 68.54 | 2.26 | 80.3 | 2.55 | 79.91 | 4.29 |
| Qwen2.5-Math-1.5B-Instruct | 78.24 | 1.59 | 82.9 | 3.19 | 81.96 | 5.31 |
| Qwen2.5-7B-Instruct | 91.13 | 3.23 | 82.3 | 3.43 | 79.68 | 4.42 |

## 4.3 MODEL GPU MEMORY USAGE

Table 9 and Table 10 show the GPU memory usage (in Gigabytes) of the proposed method with LLaDA-8B-Instruct and Dream-7B-Instruct as diffusion models, respectively. The GPU memory is recorded directly via `torch.cuda.max_memory_allocated()`. As shown in the tables, the proposed guided unmasking scheme introduces minimal memory overhead compared to the combined memory consumption of the single auto-regressive and baseline diffusion models.

Table 5: LLaDA-8B-Instruct Guided Diffusion Memory Usage (GB).

| AR Guider | DLM (GB) | Guide (GB) | Total (GB) |
|---|---|---|---|
| Qwen2.5-1.5B-Instruct | 20.7 | 4.1 | 24.8 |
| Qwen2.5-7B-Instruct | | 17.3 | 38.0 |

Table 6: Dream-Instruct-7B Guided Diffusion Memory Usage (GB).

| AR Guider | DLM (GB) | Guide (GB) | Total (GB) |
|---|---|---|---|
| Qwen2.5-1.5B-Instruct | 14.6 | 4.1 | 18.7 |
| Qwen2.5-7B-Instruct | | 17.3 | 31.9 |

## 5 CONCLUSION AND DISCUSSION

This paper presents a novel method designed for diffusion language models. The proposed `FreeCache` enables training-free caching with an off-the-shelf diffusion model. On top of that, `Guided Diffusion` facilitates the quality of diffusion by guiding the direct diffusion process with outstanding speedup. Different from prior works that relies on training or additional finetuning effort, the proposed method enables the end-to-end speedup by using the off-the-shelf diffusion model only. Compared to the baseline diffusion language models, the proposed method achieves an average of $12.14\times$ speedup across different reasoning tasks with negligible accuracy degradation. More importantly, for the first time, our method enables diffusion model to achieve the comparable performance compared to the autoregressive language models while maintaining the similar accuracy.

Despite these advantages, certain trade-offs remain: `FreeCache` trades accuracy and memory for speed, as its KV approximation introduces small but unavoidable errors and higher memory use. `Guided Diffusion` adds system complexity, requiring an extra AR guider, and the guider's alignment does impact the effectiveness of the method.

Additionally, our evaluation is based on a limited set of available large-scale DLMs. While we demonstrate generality across both Dream-7B (adapted from an AR model) and LLaDA-8B (trained from scratch), the landscape of billion-parameter DLMs remains small. The effectiveness of our methods on future DLM architectures is an area for future work.

## 6 ACKNOWLEDGEMENT

This work is supported in part by NSF grant CFF-2326608, the CoCoSys Center in JUMP 2.0, an SRC program sponsored by DARPA, and in-kind support from Google.

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

## A    LLM Usage

We used large language models, specifically ChatGPT (GPT-4, OpenAI) and Gemini (Google), in limited ways during this work. First, we used them to improve grammar, phrasing, and clarity of exposition, but not to generate substantive technical content. Second, we occasionally consulted them to help identify potentially relevant related work, while ensuring that all citations were independently checked and verified by the authors. Third, we used them at early stages as brainstorming aids to generate alternative perspectives or potential problem framings. All research directions, methods, and experiments were conceived, designed, and validated by the authors. All outputs from large language models were critically evaluated before inclusion, and the intellectual contributions of the paper are entirely the authors' own.

## B    Supplementary Material

### B.1    Code & Data Release Plan

The implementation of our methods are available at `https://github.com/ZhanqiuHu/flash-dlm-experimental`.

## C    Additional Experimental Results

### C.1    Additional Experimental Results of LLaDA-8B-Instruct

Besides the comprehensive evaluation with the SoTA Dream-v0-Instruct-7B diffusion model, we further validate the effectiveness of the proposed method with LLaDA-8B-Instruct Nie et al. (2025). As presented in Table 7, the proposed `FreeCache` and guided unmasking have demonstrated the consistent speedup on the GSM8K dataset with 8-shot chain-of-thought (CoT).

The baseline LLaDA-8B-Instruct Nie et al. (2025) can optionally support semi-autoregressive unmasking by dividing the sequence into several blocks and unmasking them from left to right. The user-defined hyperparameter `block length` defines the size of the block Nie et al. (2025). However, this does not make the generation autoregressive, and the model still processes the full sequence length at each denoising step.

In our proposed guided diffusion method, `block length` is not used or needed since the diffusion model generates all draft tokens at each step, while the AR model is used to guide the unmasking process. In other words, the "effective block length" of the guided unmasking is the length of all the diffused draft tokens at each step of drafting.

### C.2    Additional Experimental Results on Dream-v0-Instruct-7B

In addition to the benchmarks reported in Table 3, Table 8 shows HellaSwag (5-shot) results for Dream-v0-Instruct-7B with FreeCache and guided diffusion, highlighting substantial speedup and accuracy gains over the baseline.

Table 8: HellaSwag accuracy, latency, and speedup relative to the Dream-v0-Instruct-7B Baseline.

| Model / Variant | Accuracy (%) | Latency (s) | Speedup |
|---|---|---|---|
| Baseline | 73.30 | 21.99 | **1.00×** |
| FreeCache + Qwen2.5-1.5B-Instruct Guided (This work) | **76.63** | 2.46 | **8.94×** |
| FreeCache + Qwen2.5-7B-Instruct Guided (This work) | **76.84** | 3.31 | **6.64×** |

### C.3    Model GPU Memory Usage

Table 9 and Table 10 show the GPU memory usage (in gigabytes) of the proposed method with LLaDA-8B-Instruct and Dream-v0-Instruct-7B as diffusion models, respectively. The GPU memory is recorded directly via `torch.cuda.max_memory_allocated()`. As shown in the tables, the

Table 7: GSM8K (8-shot) accuracy, latency, and speedup for LLaDA-8B-Instruct with block sizes 32 and 64 and guided diffusion with Qwen2.5-Instruct models. Speedups are relative to the corresponding LLaDA baseline latency.

| Model / Setting | Details | Accuracy (%) | Latency (s) | Speedup |
|---|---|---|---|---|
| *LLaDA Baselines* | | | | |
| **Block Length = 32** | | | | |
| Baseline | | 81.52 | 57.02 | **1.00** $\times$ |
| `FreeCache` (This work) | | 79.76 | 9.20 | **6.20** $\times$ |
| **Block Length = 64** | | | | |
| Baseline | | 79.30 | 56.89 | **1.00** $\times$ |
| `FreeCache` (This work) | | 77.18 | 9.02 | **6.32** $\times$ |
| *Guided Diffusion (Qwen2.5-Instruct) with* `FreeCache` | | | | |
| Qwen2.5-1.5B-Instruct | Top-1 Match | 79.91 | 4.29 | **13.29** $\times$ |
| Qwen2.5-3B-Instruct | Top-1 Match | 79.53 | 5.16 | **11.05** $\times$ |
| Qwen2.5-7B-Instruct | Top-1 Match | 79.30 | 5.21 | **10.94** $\times$ |
| Qwen2.5-1.5B-Instruct | Top-2 Match | 78.85 | 4.40 | **12.96** $\times$ |
| Qwen2.5-3B-Instruct | Top-2 Match | 79.75 | 4.40 | **12.96** $\times$ |
| Qwen2.5-7B-Instruct | Top-2 Match | 79.68 | 4.50 | **12.67** $\times$ |
| Qwen2.5-1.5B-Instruct | Top-5 Match | 79.91 | 3.80 | **15.01** $\times$ |
| Qwen2.5-3B-Instruct | Top-5 Match | **80.06** | 4.29 | **13.29** $\times$ |
| Qwen2.5-7B-Instruct | Top-5 Match | 79.76 | 4.42 | **12.90** $\times$ |

proposed guided unmasking scheme introduces minimal memory overhead compared to the combined memory consumption of the single auto-regressive and baseline diffusion models.

Table 9: LLaDA-8B-Instruct Guided Diffusion Memory Usage (GB)

| AR Guider | Diff. (GB) | Guide (GB) | Total (GB) |
|---|---|---|---|
| Qwen2.5-0.5B | | 1.9 | 22.6 |
| Qwen2.5-1.5B | 20.7 | 4.1 | 24.8 |
| Qwen2.5-3B | | 6.7 | 27.4 |
| Qwen2.5-7B | | 17.3 | 38.0 |

Table 10: Dream-v0-Instruct-7B Guided Diffusion Memory Usage (GB)

| AR Guider | Diff. (GB) | Guide (GB) | Total (GB) |
|---|---|---|---|
| Qwen2.5-0.5B | | 1.9 | 16.5 |
| Qwen2.5-1.5B | 14.6 | 4.1 | 18.7 |
| Qwen2.5-3B | | 6.7 | 21.3 |
| Qwen2.5-7B | | 17.3 | 31.9 |

## C.4 IMPLEMENTATION DETAILS

The proposed method directly employs the off-the-shelf diffusion and auto-regressive large language models (for guided masking) from HuggingFace. The proposed method uses the same set of configurations for all tasks. We believe a more fine-grained hyper-parameter setting can lead to improved performance, but we keep a single set of configurations to demonstrate the generality and practicality. Table 11 shows the detailed experimental configuration of the proposed method with Dream-v0-Instruct-7B Ye et al. (2025) and LLaDA-8B-Instruct Nie et al. (2025).

**Stochastic Guided Unmasking.** Please note that during guided unmasking, we only consider the `Top-K` logits of the AR-guider model as the guidance logits $\log_a$. In addition to the deterministic guidance strategy (Algorithm 1 and Algorithm 2), if the maximum token logit of the diffusion output (of each token) is greater $\tau \times \max(\log_a)$, the token will be unmasked. Otherwise, it will remain silent, where $\tau$ is the `Guidance Confidence Threshold` (default = 0.5).

## C.5 PROMPT TEMPLATES

**Prompt Configuration for Question Answering and Reasoning.** We would like to highlight that our evaluation protocol encourages the diffusion model to generate the output tokens with the sense

Table 11: Diffusion configuration of the Dream-v0-Instruct-7B Ye et al. (2025) and LLaDA-8B-Instruct Nie et al. (2025) employed in this work.

| Setting | Dream-v0-Instruct-7B Ye et al. (2025) | LLaDA-8B-Instruct Nie et al. (2025) |
|---|---|---|
| **Configuration of FreeCache (This work)** | | |
| Block Cache Size | 256 | 256 |
| Max Output Tokens | 256 | 256 |
| Block Length Nie et al. (2025) | - | 64 |
| **Configuration of FreeCache + Guided Unmasking (This work)** | | |
| Max Output Tokens | 1024 | 1024 |
| Speculation Block Size | 32 | 32 |
| Top-K Assisted Tokens Selection | 2 | 2 |
| Guidance Confidence Threshold $\tau$ | 0.5 | 0.5 |

of reasoning for **both** math solving and basic common QA tasks (e.g., PiQA). As a result, the model achieves higher accuracy compared to the reported performance in Ye et al. (2025). To solidify the fact that we are not heavily relying on prompt engineering, we release the prompt templates as follows:

---

**ARC (-C/E):**

You are a careful reasoning assistant. A commonsense sentence is shown with a blank and **four** Answers that could fill the blank. Think step by step about which option makes the sentence most sensible, then decide.

### Sentence: <sentence>

### What you should do:
1. Briefly explain your reasoning.
2. On a **new line**, write exactly: "The final answer is [answer]" where **[answer]** is either **Answer1**, **Answer2**, **Answer3**, or **Answer4**. Do **not** output anything after that line.

---

**GPQA:**

You are a careful reasoning assistant. A question is shown with **four** possible answers. Think step by step about which option is correct, then decide.

### Question: <question>

### In your response:
1. First reason and explain your reasoning briefly before providing the final answer to the question.
2. Then at the end of your explanation, on a **new line**, write exactly: "The final answer is [answer]" where **[answer]** is one of **Answer1**, **Answer2**, **Answer3**, or **Answer4**. Do **not** output anything after that line.

---

**GSM8K:**

Given the following problem, reason and give a final answer to the problem.

Problem: <problem>

Your response should end with "The final answer is [answer]" where [answer] is just the final number to the problem.

---

**HellaSwag:**

You are a careful reasoning assistant. A commonsense sentence is shown with a blank and **four** Endings that could fill the blank. Think step by step about which option makes the sentence most sensible, then decide.

### Sentence: <sentence>

### In your response:
1. First reason and explain your reasoning briefly before answering this question.
2. Then at the end of your explanation, on a **new line**, write exactly: "The final answer is [answer]" where **[answer]** is either **Ending1**, **Ending2**, **Ending3**, or **Ending4**. Do **not** output anything after that line.

---

**PIQA:**

Below is an instruction that describes a task. Write a response that appropriately completes the request.

### Instruction: <instruction>

Your response should end with "The final answer is [answer]" where [answer] is the response to the problem.

---

**MMLU-PRO:**

The following are multiple choice questions (with answers) about a subject. Think step by step and then finish your answer with "the answer is (X)" where X is the correct letter choice.

The prompt of MMLU-PRO is adopted from the official repository of MMLU-PRO benchmark.

