# OpenReview forum: "FlashDLM: Accelerating Diffusion Language Model Inference via Efficient KV Caching and Guided Diffusion"
_ICLR.cc/2026/Conference — ICLR 2026 Poster_

### Official Review · Reviewer_fREk · 2025-10-29

**Soundness:** 3
**Presentation:** 3
**Contribution:** 1
**Rating:** 2
**Confidence:** 4

**Summary:**

The authors identify two main problems in current diffusion language models (DLMs). First, the lack of KV-caching, and second, the reduced quality when generating multiple tokens in parallel. The authors introduce (1) FreeCache, a KV-caching approximation, and (2) GuidedDiffusion, to improve the sampling speed and performance respectively.

**Strengths:**

1. The claim on line 207, that the KV projections for the clean tokens exhibit high temporal stability is well-supported, as the authors compute the similarity.
2. On reasoning tasks, the authors obtain faster generation with a small reduction in performance.
3. The paper is generally clear and well-written. Aside from the *key* edge case noted in the weaknesses, the implementation details should allow reproducibility.

**Weaknesses:**

### Major Weaknesses
1. **Overlap with previous work**. The two contributions of the authors overlap with prior work on discrete diffusion. Namely,
    - **FreeCache**: KV caching for diffusion language models has already been explored in prior work, which unfortunately is not acknowledged or compared against in this submission. See, for example, [1-2].
    - **GuidedDiffusion**: There is strong overlap with [3]. My understanding is that the main difference is that GuidedDiffusion here uses a large dLLM, whereas [3] uses a smaller one as the drafter and drafts only a few tokens. Since the AR and diffusion greedy predictions are supposed to align in this work, it seems important to compare GuidedDiffusion with [3], where Dream 7B would be the drafter and the AR model would be Qwen.

2. **Key edge case not discussed**: When the AR model and the diffusion model disagree on the next token (i.e., their argmax outputs differ for a given prompt), how is this conflict resolved? Does the AR model take precedence? This scenario is likely to be common, yet I did not see it discussed.

3. **Potentially unrealistic assumption**: The method assumes that the smaller AR model can reliably verify tokens, but if the AR model is weak, this verification should not work (see previous point).

4. **Poor argument of the shortcomings of parallel generation**: Section 3.1.2 misses the most important point: the main challenge in parallel generation is that DLLMs predict a *factorized* distribution, because joint modeling over all masked tokens is computationally infeasible and grows exponentially with vocabulary size (see [1-2]).

### Other weaknesses
1. Missing citation to relevant background work: D3PM, SEDD, MD4.

2. It is not stated how the similarity between KV projections in adjacent steps is measured (Figure 2). Is it the cosine similarity?

3. **Focus on semi-AR generation not clear form the background**: The background (lines 153-155) mentions position selection based on confidence, but it is not clearly stated that the authors use semi-autoregressive (semi-AR) decoding. Since their approach relies on an AR model to validate tokens, this should be explicitly clarified to improve reader understanding.

4. Lines 101-102: *"these methods are computationally expensive for computation"*. This should be clarified.

5. Lines 342-345: *"Unlike speculative decoding, Guided Diffusion does not require token-wise correction from the auto-regressive model. In other words, the proposed method produces output sequence without introducing the repetitive correction process, which guarantees the latency benefits obtained from the fast diffusion process."*. This paragraph sounds generic, and it is not clear what the authors mean. Guided Diffusion still uses an autoregressive model for generation, and, in my understanding, rejects tokens that are not the most likely under the AR model.

6. Lines 348-349: *"To that end, the reasoning power of the diffusion model is fully preserved, while the AR guider model simply ensures the causality of the output."*. This claim seems unjustified. What if the AR model is too weak? Then I would expect it will not be able to align its argmax with diffusion model.

[1] Ma et al., dKV-Cache: The Cache for Diffusion Language Models.

[2] Wu et al., Fast-dLLM: Training-free Acceleration of Diffusion LLM by Enabling KV Cache and Parallel Decoding.

[3] Christopher et al., Speculative Diffusion Decoding: Accelerating Language Generation through Diffusion.

[4] Xu et al., Energy-Based Diffusion Language Models for Text Generation.

[5] Lui et al., Discrete Copula Diffusion.

**Questions:**

1. Did the authors consider periodically refreshing the KV cache during sampling, rather than keeping it fixed throughout generation? If so, how does occasional refreshing impact performance?

2. **Focus on greedy decoding**: Lines 154-155, Equation 3: The decoding scheme assumes greedy token selection, whereas most prior work (e.g., MDLM, MD4) uses sampling for token selection and random position choice. Restricting to greedy decoding may be problematic, e.g. for coding or math. Why did the author decide to focus on greedy decoding only? Speculative decoding can handle stochastic sampling for example.

---

> ### Author Response · Authors · 2025-11-27
> **Response to Reviewer fREk - Part 1**
>
> ## Major Weakness:
> ### Relation to dKV-Cache [1] and Fast-dLLM [2]
>
> Thank you for bringing up these related approaches. Fast-dLLM determines its parallel decoding schedule through confidence-driven rules, which require selecting several hyperparameters that govern how aggressively the model unmasks positions. Its performance is sensitive to these choices, and the optimal setting can vary substantially across tasks. Likewise, dKV-Cache depends on heuristic scheduling decisions such as delay offsets, window sizes, and periodic refresh steps, each of which introduces additional hyperparameters and bookkeeping overhead. In both cases, achieving stable performance requires identifying good hyperparameters and maintaining nontrivial scheduling logic.
>
> Our approach follows a cleaner and more reliable design philosophy. Guided Diffusion uses an AR model as an agreement signal, providing a direct, model-based criterion for when positions are stable enough to unmask. This supports parallel generation without confidence thresholds, heuristic rules, or task-specific tuning, and the guider can be lightweight. FreeCache complements this by offering a minimal KV caching mechanism that avoids delay heuristics, window parameters, and refresh schedules, and it introduces no computation inside the diffusion loop. When combined, the two components allow efficient parallel unmasking without quality loss and do so without relying on hyperparameter tuning or recomputation.
>
> Our research first reveals the fact that the unmasking process of DLMs can be easily accelerated by simply introducing an autoregressive guider. Although the other works (e.g., Fast-dLLM) have identified the token dependency issue of DLMs, the speedup is achieved by relying on confidence scores and tuned thresholds and hyperparameters, whereas our approach provides an interpretable, straightforward scheme for direct deployment without task- and model-specific tuning.
>
> ### Comparison to Speculative Diffusion Decoding [3]
> Speculative Diffusion Decoding (SDD) [3] and our method both use a diffusion model together with an autoregressive model, but the functional roles of these components differ substantially. In SDD, the autoregressive model is the primary target model, and the diffusion model acts as a drafter whose proposals may be accepted or corrected by the AR model.
>
> In contrast, in our Guided Diffusion framework, we aim to preserve the underlying dLLM denoising trajectory while achieving the speedup. This allows us to avoid requiring an expensive target model. If to use SDD terminology, the diffusion language model (dLLM) plays both roles of the drafter and target: it proposes tokens and also serves as the final decision maker. The lightweight AR model is not used as a drafter nor a corrector. Instead, it acts as an unmasking supervisor that identifies positions whose tokens are already stable within the parallel denoising trajectory. The alignment between the two models provides a sufficient condition for early unmasking, but the AR model does not modify or override dLLM predictions. In this way, we can accelerate unmasking decisions without changing the underlying dLLM trajectory. We will highlight this distinction in the revised paper and include a clearer comparison to [3].

---

> ### Author Response · Authors · 2025-11-27
> **Response to Reviewer fREk - Part 2**
>
> ### Behaviour when the AR model and dLLM disagree
> Algorithm 1 specifies how the method behaves when there is no agreement between the AR model and the diffusion model predictions. For each denoising iteration, the algorithm identifies the longest prefix of masked positions where the dLLM and AR predictions match. When all positions mismatch, meaning the prefix length is zero, the algorithm falls back to unmasking only the first masked position, using the dLLM’s predicted token for that position. This corresponds to the “else” branch in Algorithm 1. All other positions remain masked and continue to be refined by the diffusion model in subsequent denoising steps.
> This mechanism ensures that even in the full-mismatch case, the diffusion model continues its normal denoising trajectory, and the AR model does not override or correct any token.
> Regarding the concern that the AR model must be strong
> The method does not require the AR model to produce high-quality text or to match the reasoning ability of the diffusion model. The AR model never determines token values. Its role is limited to checking whether its prediction matches the diffusion model prediction at a given position. This agreement is used only as a sufficient condition for early unmasking. The final token values always come from the diffusion model. Specifically, we directly employ an off-the-shelf AR model as the unmasking guider.
> This behavior is also supported by our experimental results. In Table 3 in the manuscript, we include an ablation that uses Qwen2.5-7B-Instruct as the guider, which has higher generation quality than Dream-7B-Instruct, to compare with FlashDLM using Qwen2.5-1.5B-Instruct, which has lower generation quality than Dream-7B-Instruct. In both cases, accuracy follows the Dream-7B-Instruct diffusion model rather than the AR guider.
> As shown in Table 4 of the original manuscript, the proposed guided diffusion algorithm maintains the original high accuracy of the diffusion language model itself, achieving outstanding speedup while surpassing the accuracy of the standalone non-powerful Qwen2.5-1.5B guider model (68.54% vs. 80.30%). The preserved accuracy of DLM and largely improved speedup implies the fact that the proposed algorithm does not require a strong guider model to achieve good speedup.
> Response Regarding Parallel Generation and Distribution Factorization
> We thank the reviewer for pointing out that this theoretical limitation should be stated more clearly, and we will incorporate a clarification of this point into Section 3.1.2 of the revised paper. Parallel diffusion decoding relies on a factorized distribution over masked positions, which limits its ability to model joint dependencies during parallel unmasking. Guided Diffusion operates on top of this structure. It prevents premature parallel unmasking in cases where the factorized distribution suggests instability by requiring agreement between the diffusion model and the AR model before committing multiple positions. This effectively reduces incoherence that arises from independent token predictions without altering the underlying diffusion process. In other words, the proposed Guided Diffusion algorithm doesn’t hinder the parallel diffusion of DLM itself.

---

> ### Author Response · Authors · 2025-11-27
> **Response to Reviewer fREk - Other Weaknesses and Questions**
>
> ## Other Weakness:
> ### Missing citation to relevant background work: D3PM, SEDD, MD4
> We will add citations to D3PM, SEDD, and MD4 in the background section and briefly describe how these works relate to discrete diffusion modeling and decoding.
>
> ### Similarity metric in Figure 2
> The similarity between KV projections in adjacent steps is measured using cosine similarity. We will state this explicitly in the main text and the figure caption.
>
> ### Clarification on Background
> Section 3.1 is meant to illustrate the baseline masked-diffusion formulation commonly used, which uses confidence-based unmasking, while our proposed method, which does not rely on confidence thresholds and instead commits positions based on an external agreement check, is presented in Section 3.2.
> Clarification on Lines 101-102
> Thank you for the suggestion. The compute bottleneck is discussed in more depth in Section 3.1.1. We will revise the sentence to:
> “These methods are compute-intensive because iterative denoising requires repeated full-sequence forward passes (see Section 3.1.1).”
> We will also add an explicit cross-reference from Lines 101–102 to Section 3.1.1.
>
> ### Clarification on Lines 342-345
> The AR guider’s token distribution is not used to correct diffusion proposals. At each denoising step, the diffusion model proposes tokens, and the AR model runs a single forward pass only to check agreement. We will revise the sentence to state explicitly: the AR guider runs only one forward pass to compute the next-token distribution for the current prefix and provide an agreement check; it does not correct or overwrite diffusion proposals.
>
> ### Clarification on Lines 348-349
> As shown in Algorithm 1, the AR distribution is used only to decide whether to unmask a position early and does not correct or overwrite diffusion outputs. In the corner case, if no position meets the agreement criterion in a step, we unmask only one next token according to the dLLM’s proposal, then proceed. The final outputs therefore follow the diffusion model’s capability instead of the AR guider.
>
> ---
>
> ## Questions:
> ### Regarding KV Cache Refreshment
> Thank you for bringing up this point. We did experiment with periodic refreshing on top of FreeCache. In our implementation, enabling fixed-interval refreshing requires only a single additional line, and we observed that it provides a small amount of quality recovery relative to using FreeCache alone.
>
> However, we also found that Guided Diffusion already improves quality by enforcing a more semantically coherent unmasking schedule. As shown in Table 3, Guided Diffusion recovers baseline accuracy and, in many cases, even surpasses it. Because of this, adding KV refreshing on top of FlashDLM provides limited additional quality benefits while introducing extra overhead. For this reason, we do not adopt periodic refreshing in the final design, though it remains compatible with the framework.
>
> ### Clarification on Compatible Sampling Methods
>
> Thank you for the suggestion. In Section 3.1, we are illustrating standard masked-diffusion notation; the use of $\arg\max$ in Eq.(3) is a simple instantiation, and it should not be a requirement. We will revise the text to state that token selection can use any sampling policy (e.g., greedy, temperature, etc), and that the subset $U_t \subseteq M_t$ may be chosen by confidence or at random. We will also replace the $\arg\max$ line with a policy-agnostic form, e.g.,
>
> $$
> \[
> x_{t-1}[i] \sim \text{sample}\bigl(\mathrm{softmax}(z_t[i])\bigr) \quad \text{for } i \in U_t,
> \]
> $$
>
> where $S$ denotes the chosen selection policy, we will also update Algorithm 1 and note that our later Guided Diffusion procedure is compatible with any such proposal policy and position-selection rule.
>
>
> [1] Ma et al., dKV-Cache: The Cache for Diffusion Language Models.
>
> [2] Wu et al., Fast-dLLM: Training-free Acceleration of Diffusion LLM by Enabling KV Cache and Parallel Decoding.
>
> [3] Christopher et al., Speculative Diffusion Decoding: Accelerating Language Generation through Diffusion.

---

### Official Review · Reviewer_rtU8 · 2025-10-30

**Soundness:** 3
**Presentation:** 3
**Contribution:** 3
**Rating:** 6
**Confidence:** 2

**Summary:**

The authors propose a novel, training-free method to accelerate Diffusion Language Model (DLM) inference. The approach consists of two key techniques: 1) **FreeCache**, an efficient KV cache compression strategy that reduces computational overhead, and 2) **Guided Diffusion**, a collaborative workflow that uses a lightweight autoregressive (AR) model to guide the DLM's token unmasking process. The authors validate their method through comprehensive evaluations on various downstream tasks, demonstrating significant speedups with minimal accuracy degradation. The reported performance improvements are substantial and promising.

**Strengths:**

- The paper is clearly written, with a concise and logical explanation of the motivations, insights, and methodological details.
- The experimental validation is  thorough, including both accuracy benchmarks and detailed inference latency measurements. The demonstrated inference speedup is impressive and a strong contribution.

**Weaknesses:**

- While the overall inference improvement is significant, a detailed breakdown of the contribution from each component (FreeCache vs. Guided Diffusion) would be beneficial. An ablation study showing the speedup and accuracy impact of each technique individually would help readers understand their relative importance.
- The AR-guided strategy is interesting. However, the analysis could be strengthened by including a discussion or experiment quantifying the inference overhead introduced by the AR model itself. A theoretical or empirical analysis of the computational cost of the guidance step would provide a more complete picture of the net performance gain.

**Questions:**

The paper mentions partitioning the sequence into "fixed-size blocks." How was this block size determined? what is the sensitivity of the results to this choice?

---

> ### Author Response · Authors · 2025-11-27
> **Response to Reviewer rtU8**
>
> ## Weakness
>
> ### Component-wise ablation: FreeCache vs. Guided Diffusion
> From Table 3, we can infer the separated effects. FreeCache introduces about 2-4x speedup with accuracy changes typically within a few points. On top of that, Guided Diffusion with a 1.5B guider adds an additional 1.3-9.8x over FreeCache (average 3.6x across tasks) and recovers accuracy to baseline or better on most benchmarks. Guided Diffusion with a 7B guider is included as an ablation to show that quality is governed by the diffusion model rather than the guider; its additional speed gains are smaller in practice because the 7B forward pass is more expensive than the 1.5B. Although magnitudes vary by benchmark, the pattern is consistent: FreeCache reduces per-step diffusion cost, Guided Diffusion reduces the number of steps, and a smaller guider is a reasonable default since guider size mainly affects runtime, not final quality.
> Per-step runtime breakdown with AR guider
> Thank you for the suggestion. Below, we quantify the average latency breakdown of one denoising step using FlashDLM with a 1.5B guider on top of Dream-7B-Instruct. Diffusion remains the dominant cost across datasets, with AR forward about 15-20% of total time and verification about 1-3%.
>
> **Runtime breakdown (ms; percentages are share of total)**
> | DATASET | DIFFUSION PREDICTION (ms) | AR FORWARD (ms) | VERIFICATION (ms) | TOTAL (ms) |
> |---------|----------------------------|------------------|--------------------|-------------|
> | GPQA    | 40.0 (77.5%)              | 10.2 (19.7%)    | 1.4 (2.8%)        | 51.6        |
> | GSM8K   | 61.9 (81.0%)              | 13.2 (17.3%)    | 1.2 (1.6%)        | 76.4        |
> | MMLU-PRO    | 57.8 (80.3%)              | 13.2 (18.4%)    | 1.0 (1.4%)        | 72.0        |
> | PIQA    | 54.0 (83.2%)              | 9.8 (15.2%)     | 1.1 (1.7%)        | 64.9        |
>
> ---
>
> ## Questions
> ### Block size determination and sensitivity
>
> Thanks for the question. In the current submission, we use a user-defined fixed block size for simplicity and to keep FreeCache easy to integrate with existing DLM implementations. We also stay conservative and choose a relatively large block size so that caching is not overly aggressive. In general, a larger block size performs less aggressive caching and therefore reduces the approximation loss introduced by freezing earlier KV blocks. However, the exact tradeoff is not determined by the caching approximation alone. The block size also impacts and influences the unmasking schedule of the diffusion process, since it affects how often later denoising steps revisit earlier positions. This interaction makes the choice of block size more complex than a simple latency-accuracy curve.
>
> Some hyperparameter tuning may be needed across tasks, and we view the development of adaptive or learned block-size policies as an interesting extension. Future work could also explore caching-aware or schedule-aware training so the model learns KV dynamics and unmasking behavior that are more compatible with caching.
>
> We add the following empirical results on GSM8K and GPQA for reference and comparison, using a maximum output length of 256 and varying block sizes. We observe that latency tends to increase for larger block sizes because the active window remains larger for more of the denoising process. However, accuracy trends differ across benchmarks: for GSM8K, accuracy is relatively flat and peaks around moderate block sizes, while for GPQA, accuracy improves noticeably as block size increases. This reflects differences in how sensitive each task is to block size changes.
>
> Dataset: GSM8K (8-shot)
>
> | BLOCK SIZE | ACCURACY (%) | LATENCY |
> |------------|--------------|---------|
> | 8          | 77.55        | 4.53    |
> | 16         | 78.77        | 4.65    |
> | 32         | 78.77        | 4.88    |
> | 64         | 79.30        | 5.27    |
> | 128        | 78.16        | 6.11    |
> | 256        | 77.93        | 7.95    |
>
> Dataset: GPQA
>
> | BLOCK SIZE | ACCURACY (%) | LATENCY |
> |------------|--------------|---------|
> | 16         | 33.93        | 7.53    |
> | 32         | 35.94        | 7.80    |
> | 64         | 38.17        | 7.95    |
> | 128        | 39.29        | 9.82    |
> | 256        | 43.30        | 9.91    |

---

### Official Review · Reviewer_vA4B · 2025-10-31

**Soundness:** 3
**Presentation:** 3
**Contribution:** 3
**Rating:** 6
**Confidence:** 3

**Summary:**

The paper presents FlashDLM, an inference-time framework to speed up diffusion language models without extra training. It has two parts. FreeCache observes that the Key/Value projections of finalized tokens change very little across denoising steps and reuses those projections, so later steps only recompute attention for the unfinished suffix instead of the full sequence. Guided Diffusion uses a small autoregressive model to check the DLM’s parallel token proposals and confirm the longest common prefix, which reduces the number of denoising steps and improves coherence.

**Strengths:**

1) The method is training-free.
2) The experiments connect the stability observation to a concrete algorithm and show clear improvements in latency with small or no drops in accuracy.
3) The paper motivates FreeCache with a figure that shows stability across steps and positions. The description of block partitioning, active window recomputation, and progressive freezing is easy to follow.
4) The work targets a key bottleneck for DLMs in long-context reasoning: repeated full-sequence passes and parallel decoding incoherence.

**Weaknesses:**

1) The study centers on Dream-7B-Instruct and LLaDA-8B-Instruct with reasoning benchmarks. It is unclear how FreeCache scales to much larger DLMs or to domains such as dialogue safety, coding, or open-ended writing, where coherence needs may differ. Adding more domains or larger models would strengthen external validity.
2) Guided Diffusion depends on a longest-prefix acceptance rule with top-k AR logits and a confidence threshold τ. A deeper ablation on k, τ, and different AR guiders would clarify robustness and failure modes.

**Questions:**

1) Which block sizes and shrink schedules work best across tasks? Could a dynamic policy that adapts block size based on local uncertainty or AR-agreement rate further reduce recomputation?
2) Beyond the current longest-prefix rule with top-k and threshold τ, have you tried alternatives such as per-token margin checks, KL limits, or short rollback windows, and how do they affect step count, latency, and accuracy?

---

> ### Author Response · Authors · 2025-11-27
> **Response to Reviewer vA4B - Part 1**
>
> ## Weakness
> ### Scope across models and domains
> We chose LLaDA-8B and Dream-7B because they are commonly used and openly released diffusion language models with weights and inference code. LLaDA is trained from scratch under a masked-diffusion objective, while Dream 7B is trained as a discrete diffusion language model after AR-based initialization. For both models and on the benchmarks reported in Table 3, we observe the same KV stability trend.
>
> It is possible that future architectures or tasks exhibit different stability characteristics or agreement rates. In the worst case, accuracy recovery could require additional fine-tuning, for example, applying semi-autoregressive masking during training to calibrate stability or commit decisions.
>
> ### On the longest-prefix rule, top-k, and $\tau$
> Thanks for bringing this up. We suggest to be conservative, i.e., choosing k <= 3 and tau >= 0.5, to both preserve quality and maintain agreement rate. Empirically, we observe that larger top-k (for example, k > 5) tends to lower agreement between the diffusion model and the AR guider, which reduces generation quality and often does not improve end-to-end inference.
>
> For $\tau$, we introduced this to accept near ties rather than only strict top-1 matches. If the AR guider assigns a slightly higher probability to its top-1 token than to the top-2 token, and the diffusion model proposes the top-2 token, we prefer not to miss that case. Similarly, if $\tau$ is too small (for example, may accept top-2 even when it is clearly worse than top-1), we see a drop in quality. Therefore, we also want to avoid accepting weak alternative tokens to preserve quality.

---

> ### Author Response · Authors · 2025-11-27
> **Response to Reviewer vA4B - Part 2**
>
> ## Questions
>
> ### Block sizes and shrink schedules
> Thanks for the question. In the current submission, we use a user-defined fixed block size for simplicity and to keep FreeCache easy to integrate with existing DLM implementations. In general, choosing a larger block size means performing less aggressive caching, which can reduce the approximation loss introduced by freezing earlier KV blocks. However, the exact tradeoff is not determined by the caching approximation alone. It also depends on how the block size impacts and influences the unmasking schedule of the diffusion process. This interaction makes the choice of block size more complex than a simple latency-accuracy curve.
>
> We agree that adaptive strategies are a promising next step. Potential directions include dynamically adjusting block size based on local KV stability, diffusion uncertainty, or the AR-agreement rate from Guided Diffusion. Beyond purely inference-time heuristics, one could also explore cache-aware or schedule-aware training or finetuning so that the model learns to maintain greater KV stability or to structure its unmasking behavior in ways that better support caching. We will include a brief discussion of these extensions in the revised paper.
>
> We also add the following empirical results on GSM8K and GPQA for reference and comparison, using a maximum output length of 256 and varying block sizes. We observe that latency tends to increase for larger block sizes because the active window remains larger for more of the denoising process. However, accuracy trends differ across benchmarks: for GSM8K, accuracy is relatively flat and peaks around moderate block sizes, while for GPQA, accuracy improves noticeably as block size increases. This reflects differences in how sensitive each task is to block size changes.
>
> Dataset: GSM8K (8-shot)
>
> | BLOCK SIZE | ACCURACY (%) | LATENCY |
> |------------|--------------|---------|
> | 8          | 77.55        | 4.53    |
> | 16         | 78.77        | 4.65    |
> | 32         | 78.77        | 4.88    |
> | 64         | 79.30        | 5.27    |
> | 128        | 78.16        | 6.11    |
> | 256        | 77.93        | 7.95    |
>
> Dataset: GPQA
>
> | BLOCK SIZE | ACCURACY (%) | LATENCY |
> |------------|--------------|---------|
> | 16         | 33.93        | 7.53    |
> | 32         | 35.94        | 7.80    |
> | 64         | 38.17        | 7.95    |
> | 128        | 39.29        | 9.82    |
> | 256        | 43.30        | 9.91    |
>
>
> ### Alternatives to the longest-prefix rule
> Thanks for the suggestion. Our relaxation already captures the core idea of a per-token margin check. The use of the confidence threshold $\tau$ allows accepting a DLM token when its AR logit is close to the AR model's top-1 logit, so near-tie cases are not rejected outright. This plays a similar role to margin-based acceptance while keeping the rule simple.
>
> We have not explored other alternatives, such as KL-bounded acceptance or rollback mechanisms, in our current experiments. Our design goal for Guided Diffusion is to keep the mechanism simple and to avoid additional operations that introduce overhead or compromise the parallelism advantage of diffusion. The longest-prefix rule, together with our use of top-k relaxation and the $\tau$ near-tie criterion, satisfies this design constraint while still offering a conservative safeguard for semantic coherence.
>
> That said, the suggestions are well aligned with the underlying objective. Margin-based tests or KL-bounded acceptance could provide finer control over how aggressively we trust the DLM draft, and short rollback windows may help in cases where local disagreements spike. These ideas fit naturally into our framework, and we will incorporate a discussion of them in the revised paper as possible extensions.

---

### Official Review · Reviewer_GXxp · 2025-11-01

**Soundness:** 2
**Presentation:** 3
**Contribution:** 2
**Rating:** 6
**Confidence:** 2

**Summary:**

### **Paper Summary**
Diffusion Language Models (DLMs) offer advantages like **parallel token generation** and **inherent bidirectionality** over autoregressive (AR) LLMs, but suffer from slow inference—their iterative denoising requires multiple full-sequence forward passes, leading to high latency. Additionally, parallel generation causes token incoherence, and reducing denoising steps degrades output quality.  To address these issues, the paper proposes **FreeCache** and **Guided Diffusion** techniques under the FlashDLM framework. **FreeCache** is  a training-free KV caching technique for DLMs: It exploits the temporal stability of KV projections in unmasked tokens to reduce redundant computations, enabling efficient long-context inference without fine-tuning. **Guided Diffusion** is a training-free cross-model guidance method: By using a lightweight AR model to validate DLM draft tokens, it resolves token incoherence and drastically reduces denoising iterations while preserving the DLM’s reasoning power.  Extensive experiments on benchmarks show the combined methods deliver an **average 12.14× end-to-end speedup** with negligible accuracy degradation.

**Strengths:**

### **Strengths of the Proposed Method**
1. **Training-Free Design for Off-the-Shelf Deployment**
The core techniques (FreeCache and Guided Diffusion) require no additional training, fine-tuning, or dedicated calibration runs—they directly accelerate pre-trained DLMs "off the shelf". This avoids the overhead of retraining large models and significantly lowers the barrier to practical adoption.

2. **FreeCache: Targeted KV Caching to Cut Redundant Computation**
FreeCache leverages a key insight—KV projections of "clean" (unmasked) tokens exhibit high temporal stability across denoising steps—to dynamically shrink the active computation window. By freezing KV projections of completed token blocks and only recomputing for pending blocks, it eliminates redundant full-sequence calculations inherent to vanilla DLMs.

3. **Guided Diffusion Resolves Parallel Generation Incoherence**
Guided Diffusion addresses DLM’s critical token incoherence issue by using a lightweight AR model as a "coherence supervisor". It only unmask tokens where the DLM’s draft predictions match the AR model’s outputs, ensuring semantic consistency without sacrificing parallel generation advantages. Unlike heuristic sampling methods that suffer severe accuracy loss with fewer steps, Guided Diffusion recovers accuracy even with aggressive denoising reductions.

4. **Breakthrough Latency Parity with Autoregressive (AR) Models**
For the first time, the method enables DLMs to match or outperform same-sized AR models in latency while preserving accuracy. This bridges the latency gap that previously restricted DLM adoption in high-throughput scenarios.

5. **Low Memory Overhead for Practical Deployment**
Guided Diffusion introduces minimal additional memory usage when combining DLMs and AR guiders. This efficiency makes the framework suitable for edge or resource-constrained environments.

**Weaknesses:**

### **Weaknesses and Restrictions**
1. **Limited Validation Across DLM Architectures and Scales**
The method’s experiments are restricted to only two DLMs: Dream-7B-Instruct and LLaDA-8B-Instruct. It lacks testing on larger DLM variants (e.g., 13B/34B parameter DLMs) or domain-specific DLMs, making it unclear whether FreeCache’s KV stability assumption or Guided Diffusion’s AR supervision generalizes to more complex or specialized DLM structures.

2. **Sensitivity to the Quality and Domain Alignment of AR Guiders**
Guided Diffusion relies entirely on a lightweight AR model to ensure token coherence, but the paper provides limited analysis of how AR guider quality or domain mismatch impacts performance.

3. **Fixed Block Size in FreeCache Lacks Adaptive Optimization**
FreeCache uses a fixed block size for all tasks and models, with no analysis of how block size impacts latency or accuracy. For short-sequence tasks, a 256-token block may lead to underutilization of the active window. For ultra-long sequences, a fixed block size could prevent optimal window shrinking. The lack of adaptive block size tuning limits FreeCache’s efficiency across sequence lengths and task types.

5. **Insufficient Comparison with State-of-the-Art DLM Acceleration Methods**
While the method compares against vanilla DLMs and AR models, it lacks head-to-head evaluation with other state-of-the-art DLM acceleration techniques.

**Questions:**

### **Questions and Suggestions for Authors**

1. **Can FLASHDLM Maintain Performance on Ultra-Long Contexts?**
The paper claims support for long-context generation (>1024 tokens) but only tests up to 1024 tokens. For ultra-long contexts, FreeCache’s frozen KV blocks may accumulate to cause memory bottlenecks, and Guided Diffusion’s matching ratio could drop due to increased semantic drift. Does FLASHDLM’s latency-accuracy tradeoff hold for these extended sequences, and are there optimizations needed to support them?

2. **Implement Adaptive Block Size for FreeCache to Improve Task Adaptability**
Given the limitations of a fixed 256-token block size, the authors should design an adaptive block size mechanism. For instance, use smaller blocks for short sequences to reduce redundant computation, and larger blocks for long contexts to minimize window-shrinking overhead.

3. **Evaluate Guided Diffusion with Low-Resource and Domain-Specific AR Guiders**
To enhance practicality, the authors should test Guided Diffusion with low-resource AR models and domain-mismatched guiders.

---

> ### Author Response · Authors · 2025-11-27
> **Response to Reviewer GXxp - Part 1**
>
> We thank the reviewer for the detailed and constructive feedback, which helped us clarify several important points.
>
> ## Weakness
> ### Limited Validation Across DLM Architectures and Scales
>
> Thank you for pointing this out. Publicly available diffusion language models are still limited in scale. To our knowledge, Dream-7B-Instruct and LLaDA-8B-Instruct are among the largest and most commonly used open-source DLMs with released weights and inference code, which constrains the range of models that can be evaluated at submission time. LLaDA is trained from scratch under a masked-diffusion objective, while Dream 7B is trained as a discrete diffusion language model after AR-based initialization. We observe consistent effectiveness of our approaches on both of these models.
>
> We acknowledge that there are emerging multimodal and domain-specific diffusion models, and evaluating additional architectures and domains would be valuable. We will clarify this limitation and add a short discussion in the revision.
>
> ### Sensitivity to AR Guider Quality or Domain Alignment
> Thank you for raising this point. The empirical results in Table 4 illustrate that Guided Diffusion produces similar accuracy across AR guiders of very different strengths. For example, using Qwen2.5-1.5B-Instruct, Qwen2.5-Math-1.5B-Instruct, or Qwen2.5-7B-Instruct, the final accuracy remains close to the capability of the diffusion model rather than the guider itself, even though these AR models differ greatly in their standalone AR accuracy. This indicates that the quality of Guided Diffusion is not determined by the AR model. The reason is that the AR model is not used to generate or correct tokens. It only provides an agreement signal that indicates when a position is stable enough to unmask early. However, similar to other unmasking strategies, the quality of the final generation can vary slightly depending on the unmasking schedule.
>
> ### Fixed Block Size in FreeCache Lacks Adaptive Optimization
>
> Thank you for raising this point. In the current submission, FreeCache relies on a user-defined block size. We choose a relatively large block size to stay conservative, since larger blocks reduce the risk of introducing noticeable approximation loss when earlier KV states are frozen. However, the tradeoff is not determined by the caching approximation alone. It also depends on how the block size interacts with the denoising process. This coupling makes the choice of block size more involved than a single speed-accuracy curve, and different tasks can respond differently.
>
> We agree that adaptive or learned block-size schedules would improve flexibility. Although we do not implement such mechanisms here, FreeCache is compatible with dynamic policies based on signals such as KV stability, model uncertainty, or agreement statistics from Guided Diffusion. We will note these possible extensions in the revised version.
>
> For reference, we provide a small block-size study at the end of this response. In summary, latency increases steadily with larger block sizes because the active window remains wider for more steps. Accuracy, however, diverges across tasks: GSM8K remains relatively stable and peaks around moderate block sizes, while GPQA improves consistently as block size increases.
>
>
> Dataset: GSM8K (8-shot)
>
> | BLOCK SIZE | ACCURACY (%) | LATENCY |
> |------------|--------------|---------|
> | 8          | 77.55        | 4.53    |
> | 16         | 78.77        | 4.65    |
> | 32         | 78.77        | 4.88    |
> | 64         | 79.30        | 5.27    |
> | 128        | 78.16        | 6.11    |
> | 256        | 77.93        | 7.95    |
>
> Dataset: GPQA
>
> | BLOCK SIZE | ACCURACY (%) | LATENCY |
> |------------|--------------|---------|
> | 16         | 33.93        | 7.53    |
> | 32         | 35.94        | 7.80    |
> | 64         | 38.17        | 7.95    |
> | 128        | 39.29        | 9.82    |
> | 256        | 43.30        | 9.91    |

---

> ### Author Response · Authors · 2025-11-27
> **Response to Reviewer GXxp - Part 2**
>
> ### Insufficient Comparison with State-of-the-Art DLM Acceleration Methods
> Thank you for pointing this out. Several concurrent works on DLM acceleration emerged around the same time as our manuscript. Although full head-to-head comparisons were not feasible at submission time, we will add a clear discussion of these approaches and how our design choices differ.
> Concurrent KV-reuse methods typically accelerate denoising by reusing intermediate KV states and then applying partial or full recomputation at selected steps to correct drift and recover accuracy. Separately, concurrent parallel-decoding approaches introduce confidence- or threshold-based unmasking rules to enable broader parallelism, which requires careful hyperparameter tuning to balance quality and speed.
> Our approach is driven by a cleaner and more reliable design philosophy. Guided Diffusion enables parallel unmasking without any threshold tuning by using model agreement as the sole unmasking signal. Our KV design is also motivated by the empirical observation that clean-token KV projections are temporally stable, but FreeCache deliberately avoids recomputation and, when used alone, introduces a modest accuracy drop. Accuracy is recovered when FreeCache is combined with Guided Diffusion, which compensates for the approximation error and restores quality close to the baseline without recomputation and without tuning. In this combined setting, we also tested periodic recomputation and observed only limited quality gains with noticeable latency cost, so we did not adopt recomputation.
>
> ---
>
> ## Questions
> ### Can FLASHDLM maintain performance on ultra-long contexts?
>
> Thank you for bringing up the point. At very long lengths, methods that retain KV states can face memory pressure, and FreeCache is subject to the same general constraint. Practical mitigations exist and can be adapted from AR inference, including sliding or truncated KV windows, eviction of distant frozen blocks, quantized or mixed-precision KV storage, and paging or offloading inactive KV blocks. These options do not alter the core mechanism of FreeCache.
>
> Current open diffusion language models are not commonly evaluated for ultra-long sequences because they seldom generate very long answers by themselves. However, as RLHF-style preference optimization for DLMs matures, longer free-form generations may become practical, at which point it will be valuable to analyze ultra-long behavior and resolve any issues that may arise.
> Implement Adaptive Block Size for FreeCache to Improve Task Adaptability
> Thank you for the suggestion. We agree that an adaptive block-size mechanism could further improve efficiency. Although we did not explore adaptive policies in this submission, FreeCache is compatible with any block-size schedule, including dynamic ones. Potential extensions include adapting block size based on local KV stability, diffusion uncertainty, or the agreement rate from Guided Diffusion. More advanced options include caching-aware or schedule-aware training or finetuning so the model naturally adopts KV patterns and unmasking behavior that better support caching. We will add this discussion to the revision.
> Evaluate Guided Diffusion with Low-Resource and Domain-Specific AR Guiders
> As discussed in the weakness section above, we evaluated guiders of very different strengths and domains and observed that Guided Diffusion is not sensitive to guider quality because the guider only provides an agreement signal. We will briefly note this observation in the revision and mention that exploring additional guider types is a natural extension.

---

### Author Response · Authors · 2025-12-03
**Summary of Our Contributions and Rebuttal Clarifications**

Dear AC,

We appreciate all the valuable feedback and comments from reviewers. Before moving to the next phase, we summarize the contribution of our work and our effort during the rebuttal.

## Contribution:
The goal of our work is clear and novel: **Minimizing the efficiency gap (wall-clock) between diffusion language models (DLMs) and autoregressive models (AR) without extra fine-tuning or heuristic hyperparameter search.** We believe "plug-in and play” acceleration is crucial nowadays for consistent speedup across various tasks (reasoning and non-reasoning).

To achieve this goal, we propose **1) FreeCache** and **2) Guided Diffusion**, achieving 12.14$\times$ average speedup *across diverse tasks*. Based on the observations (both LlaDA and Dream DLMs) that cleaned and unmasked tokens stay stable across diffusion steps, FreeCache directly accelerates diffusion with minimal accuracy loss. On top of that, Guided Diffusion further improves speed by using a lightweight AR model (e.g., Qwen2.5-1.5B) to guide which tokens are unmasked while still letting the DLM perform the denoising. Different from speculative decoding (for AR models) and Speculative Diffusion models , the proposed Guided Diffusion only uses the lightweight AR model to guide which token should be unmasked, we do not repeatedly correct DLM outputs; thus, the repetitive correction overhead (e.g., SDD and AR-based Speculative Decoding) is removed, and the DLM acts as both target and draft models.

Our findings are meaningful for the DLM research community. First, both proposed methods are directly employed based on the off-the-shelf DLM and AR guider models. , and DLM latency becomes comparable to AR models even on a plain HuggingFace backend, as demonstrated in our anonymously released code. We believe the latency of the proposed method can be improved even further with the DLM-specialized CUDA kernels with maximized parallelism, which will be investigated in future work. Second, unlike Fast-dLLMv2 or TiDAR, our method requires no additional training or heuristic parameter selections, providing user-friendly acceleration. Finally, we show that DLMs and AR models can be effectively combined despite architectural differences.

## Summary of our rebuttal
We have prepared detailed responses to address all the questions. Reviewer GXxp questioned block-size tuning and AR-guider quality. We clarified the diffusion process and showed robustness via block-size sweeping. Reviewer vA4B questioned whether the feasibility of FreeCache is mainly determined by the long-prefix tasks. As shown in the supplementary material, we intentionally ask the model to provide a holistic reasoning process for multiple-choice questions (e.g., PiQA). By intentionally encouraging the proposed FlashDLM system to generate long-context output, we prove that the impact of prefix lengths is minimal, and the proposed method maintains almost lossless accuracy while having outstanding speedup. Reviewer rtUB asked about block sizes and AR-guider overhead; empirically, AR-guider forward passes account for only 15–20% of runtime and verification 1–3%, while FreeCache remains robust across block sizes.

Reviewer fREk compared FreeCache with dKV-Cache and Fast-dLLM. Although both of these prior works exploit the plausibility of introducing caching to DLM, Fast-dLLM requires intricate and task-specific hyperparameter tuning to govern how aggressively the DLM model unmasks the positions. Likewise, dKV-Cache introduces heuristic scheduling decisions such as delay offsets, window sizes, and periodic refresh steps. On the contrary, the proposed FreeCache method shows strong robustness across various block sizes in terms of block size of FreeCache (Table below) and guider model selection of Guided Diffusion (Table 4-6 of the original manuscript).

In addition to the questions regarding the Caching mechanism, Reviewer fREk also asks about the difference between our proposed Guided Diffusion with Speculative Diffusion Decoding (SDD). We would like to highlight the fact that SDD uses an AR model as the **target model**, while the diffusion model is employed for **drafting**, just like the classic speculative decoding for AR models. On the contrary, our method instead preserves a Diffusion-dominant pipeline: a lightweight AR model only guides unmasking, performs no token-wise corrections, and the DLM remains both target and draft. Such an architectural difference between SDD and our method is critical; the proposed guided diffusion still preserves the underlying dLLM denoising trajectory while achieving the speedup, while keeping the expensive autoregressive decoding minimal. With emerging DLM-oriented AI infrastructures (e.g., SGLang-Diffusion), we believe the speedup of the proposed method can be further facilitated, which will be investigated in the future.

Finally, we appreciate all suggestions regarding figures, context, and writing, and will incorporate them into the updated manuscript.

---

### Meta-Review · Area_Chair_a2RC · 2026-01-07

**Summary:**

Based on the generally positive reviews, I recommend accepting this paper. Reviewers initially raised concerns regarding hyperparameter sensitivity (e.g., block sizes), the computational overhead of the AR guider, and potential overlap with Speculative Diffusion Decoding (SDD). The authors effectively addressed these by demonstrating robustness across block sizes and providing detailed latency breakdowns. Crucially, they clarified the structural distinction from SDD, highlighting that FlashDLM maintains the DLM as the target model rather than the AR model. The simple and training-free nature of FlashDLM, combining FreeCache and Guided Diffusion, provides a significant and practical speedup for Diffusion Language Models.

**Reviewer Concerns:**

Addressed:
1. Hyperparameter Sensitivity: The rebuttal provided empirical evidence showing that performance remains robust across different block sizes for reasoning tasks, addressing concerns from Reviewers GXxp and vA4B.
2. Computational Overhead: Reviewer rtU8's request for a detailed latency breakdown was fully addressed, showing the AR guider adds minimal overhead (15-20%).
3. Novelty & SDD Distinction: The authors successfully clarified for Reviewer fREk that unlike Speculative Diffusion Decoding, their method retains the DLM as the target/drafter while using the AR model solely for unmasking guidance.

Outstanding:
1. Concurrent Baselines: A direct head-to-head empirical comparison with concurrent acceleration methods (e.g., Fast-dLLM) is missing, though the conceptual differences were discussed.

**Reviewer Scores:**

1. Reviewer GXxp: 6 -> 6
2. Reviewer vA4B: 6 -> 6
3. Reviewer rtU8: 6 -> 6
4. Reviewer fREk: 2 -> 4

---

### Decision · Program_Chairs · 2026-01-26

Accept (Poster)